# Suspected Cases of Chlamydia-Induced Fertility Problems in Sows: What Is the Approach of Austrian Practitioners?

**DOI:** 10.3390/ani14202983

**Published:** 2024-10-16

**Authors:** Christine Unterweger, Andrea Buzanich-Ladinig

**Affiliations:** Clinical Unit for Swine Medicine, Clinical Department for Farm Animals and Food System Science, University of Veterinary Medicine Vienna, Veterinaerplatz 1, 1210 Vienna, Austria; andrea.ladinig@vetmeduni.ac.at

**Keywords:** *Chlamydia suis*, fertility disorders, questionnaire, tetracycline resistance, prevention

## Abstract

*Chlamydia* spp. are obligate intracellular bacteria mostly known to cause fertility problems and trachoma in humans but also cause a variety of diseases of major interest in animals. Since some veterinary chlamydial species are known to be zoonotic, it is important to monitor the entire chlamydial outcomes worldwide in humans and animals. Pig farmers and veterinarians are in close contact with swine chlamydia every day. What is not yet fully known is the clinical role that chlamydia plays in swine health, especially in the context of fertility problems. There are suspicions that chlamydia in the genital tract of sows may play a role in infertility, but there is currently no evidence of this. This survey was conducted among veterinarians who diagnose daily in pig farms and must make decisions regarding treatment and prevention. Practitioners are calling for significant action and support from the scientific community because their decisions on diagnosis, therapy, and prevention are often based on intuition rather than scientifically founded knowledge, due to the lack of scientifically validated guidance on control strategies including treatment length, treatment dose, and disinfection procedures. This would not only be important for providing adequate veterinary care to the sows but also in terms of the One Health principle.

## 1. Introduction

Almost everything we know so far about chlamydia in association with fertility problems in sows and boars originates from case reports from the veterinary field. In sows, these include cases of abortions, weak or stillborn piglets, mummifications, and small litter sizes as well as infertility and rebreeding [1]. The number of case descriptions is low, though. Most of the cases reported come from Europe and are relatively old [2,3,4,5,6,7]. Nevertheless, these case reports represent the basis for all the typical clinical signs listed in the current literature for genital chlamydiosis in pigs, as Koch’s postulates have not been fulfilled so far. As a consequence of this lack of knowledge from successfully conducted *C. suis* in vivo studies, there is also no gold standard for diagnostics either for direct or for indirect detection; in addition, there is no defined standard for the selection of appropriate sampling material for chlamydia detection. Accordingly, there are no established criteria for the assessment of genital chlamydiosis. Antibiotic treatment poses another big issue, as there is limited information available on the current sensitivities of chlamydial isolates to antimicrobial agents [8]. Isolation of chlamydia is only feasible by cell culture methods, which makes routine chlamydia testing and consequently susceptibility testing not an option. Nevertheless, tetracyclines are generally recommended as drugs of choice [1]. Meanwhile, it is known that for the treatment of *Chlamydia* (*C.*) *suis*, the use of tetracyclines might be counterproductive, since this is the only species of the *Chlamydiaceae* family to have naturally acquired a tetracycline resistance gene, which is part of a genomic island integrated into the chromosome [9,10]. In addition, there are no studies on the most appropriate duration of treatment or delivery route. While antimicrobial treatment destroys the reticulate bodies—the non-infectious and metabolic active intracellular stages of chlamydia—the elementary bodies—the infectious extracellular stages—can only be inactivated by using adequate disinfection protocols, which are also missing. Methods developed for controlling other chlamydial species may not be fully effective for *Chlamydia suis*, on the one hand due to the aforementioned tetracycline resistance and antibiotic regulations in livestock medicine, and on the other hand due to differences in the husbandry of herd animals in barns, usually associated with a high animal density compared to pets kept individually. There are huge knowledge gaps regarding common infection sources, infection routes, potential vectors, and infection kinetics in pig herds. Finally, unlike *C. abortus* vaccines in ruminants or *C. felis* vaccines in cats, vaccines against *C. suis*-induced clinical problems in pigs are not available [1]. This study aims to characterise the attitudes of swine practitioners in Austria, who deal with this subject on a regular basis and therefore have an obligation to farmers to find solutions for therapy and prevention of chlamydiosis in sows. The answers of this study will help researchers to understand problems which arise in the field and open new areas of investigation with the main aim of the prevention of chlamydiosis in sows.

## 2. Materials and Methods

In 2022, an anonymous survey was distributed once by email among 85 Austrian practitioners specialised in porcine health medicine and management using the free online survey tool LimeSurvey (https://www.limesurvey.org, accessed on 15 June 2022). The email addresses in the contact database of the authors’ institution were used to invite swine practitioners to fill in the questionnaire. The response percentage was calculated from the total number of veterinarians; those that did not open the email were not excluded from the calculation. The 42 respondents (49.4%) worked in the largest pig practices in Austria, mainly in the pig-dense areas of Upper Austria, Lower Austria, and Styria, and had at least five years of professional experience. Within the first question, they were informed that the data will be processed anonymously, and they all gave written consent for their data to be analysed and published. The questionnaire was delivered via email, and no advertising was made in veterinary newspapers or any other social media. The questionnaire was prepared in German and consisted of four multiple-choice questions (2–7 answers/question), one single-answer question, two open questions, and three questions with ten predefined answer options on a Likert scale (Appendix A). The questions were closed-ended, but in the multiple-choice questions, by filling in the “other” field, respondents could give their own answers not included in the listed options. It was not possible to skip questions or to view or change answers that had already been completed. With the exception of lengths of professional experience, no personal information was requested. The questions focussed on four main topics: approach to sampling, diagnostic procedures, antimicrobial usage, and prophylactic measures in case of a suspicion of genital chlamydiosis in sows. All results were reported as absolute numbers and percentages. The descriptive analysis of the responses was analysed using Microsoft Office Excel 2013 (Microsoft Excel, Microsoft Corporation, Redmond, WA, USA).

## 3. Results

More than half of the respondents (59.5%; n=26) generally consider chlamydial infections in pigs to be of little or no significance. In contrast, 40.5% (n = 17) attribute a moderate significance to chlamydia. None of the respondents (grades 8–10) recognised them as (very) important. However, the importance of chlamydial infections from the practitioners’ point of view increases as soon as fertility problems occur (Figure 1). In total, 11.9% of practitioners (n = 5) attributed great importance to chlamydial infections in farms with reproductive disorders (grade 8–10); the majority (61.9%; n = 26) described the importance as moderate (grade 4–7), while 26.19% of practitioners (n = 11) graded chlamydia in connection with fertility problems as insignificant or at least of little importance (grade 1–3).

The most frequently named symptoms were rebreeding (78.6%; n = 33)—meaning that previous mating attempts did not succeed—followed by vaginal discharge (61.9%; n = 26), abortions (52.4%; n = 14), and the occurrence of weak or stillborn piglets (16.7%; n = 7) and mummies (7.1%; n = 3). Five participants (11.9%) (“Other answers”) considered chlamydia not to play any role in fertility problems or at least had doubts about their importance, since they had never been able to detect them in sows or foetal material before (Figure 2).

The different attitudes and personal clinical experiences are also reflected in the way diagnostics are implemented. None of the respondents stated that they never initiate diagnostic testing in suspected cases, but around one-fifth (21.4%; n = 9) reported that they only rarely initiate diagnostic procedures for chlamydiosis in the case of fertility problems. In contrast, 40.5% of practitioners (n = 17) selected the answers 4 to 7 (sometimes), and 38.1% (n = 16) specified that they often to always perform diagnostic procedures in suspected cases (grades 8–10). Nevertheless, there is widespread dissatisfaction among the respondents with the currently available test methods for diagnosing chlamydia-associated fertility disorders. The questionnaire results indicate that 73.8% of respondents (n = 31) are unable to provide a reasonable diagnosis with the currently available tests, whereas 26.19% (n = 11) are satisfied with the tests available. Difficulties are reported in particular with the interpretation of antibody measurements using the complement fixation test (CFT), the most preferred screening method for genital chlamydial infections in many countries (69.1%, n = 29). In addition, PCR tests of vaginal swabs (40.5%; n = 17), abortion material (64.3%; n = 27), and the genital tract (e.g., post mortem as part of abattoir screening) (9.5%; n = 4) were chosen for molecular detection. Histopathological examinations are not requested by practitioners on a routine basis (Figure 3).

The majority (95.2%) of respondents (n = 40) use tetracyclines as the treatment of choice for chlamydia-related fertility disorders. Macrolides were mentioned as suitable antibiotics by one respondent, while three other respondents (7.14%) stated the use of other antibiotic groups without naming them. All used antibiotics as a treatment option. Antibiotic treatment is most often performed for 11–15 days (50.1% of respondents, n = 21); 26.8% (n = 11) treat the sows for 5–10 days. Some participants also use antibiotics for longer than 15 days (11.9%; n = 5) or even longer than 21 days (9.5%; n = 4). No respondent treated sows for less than five days (Figure 4). Again, a significant discrepancy arises because there are no official treatment recommendations beyond the use of tetracyclines. Additionally, a longer treatment duration might be beneficial due to the biphasic and intracellular development cycle characteristic of chlamydia. According to the practitioners, chlamydia treatment is typically administered as a whole-herd treatment, generally given orally. The high rates of tetracycline resistance in cases of *C. suis* involvement are not widely recognised.

Various prophylactic measures used to prevent chlamydia-related fertility diseases are listed in Table 1.

To enhance hygiene, cleaning and disinfection measures were specifically emphasised, along with actions such as removing faeces, maintaining dry areas, and ensuring hygiene during insemination. In order to minimise the risk of chlamydial transmission during insemination, either the boar is replaced, or artificial insemination is used. Other measures taken by individual practitioners include vaginal/uterine lavage or hormonal treatment of affected sows, although no further details were provided. In most cases, affected animals that do not respond to therapy are sent to the slaughterhouse. When antibiotics were used, no information was provided regarding the exact duration of treatment or the concentrations of the antibiotics administered.

Finally, swine practitioners suggested five key points to improve the situation regarding chlamydia-induced fertility problems:Development of improved diagnostic procedures: Respondents expressed interest in developing more reliable and sensitive testing methods to detect chlamydia and other potential fertility pathogens, such as *Leptospira* spp., accurately and at an early stage. Enhanced diagnostics would help reduce false-negative results and enable more effective early diagnosis.Identification of sources of infection: Respondents wanted a more thorough investigation into the potential sources of chlamydia infection on pig farms. This could improve understanding of how the pathogen spreads and support the development of targeted prevention and control measures.Preventive measures: In addition to hygiene improvement and rodent control, respondents were interested in other preventative measures. These include, for example, investigating the effectiveness of disinfection procedures, the use of probiotics or other feed additives to strengthen the pigs’ immune systems, and the optimisation of insemination management to reduce the risk of transmission.Research into the effects on fertility: There is interest in further studies investigating the exact effects of chlamydial infections on the fertility parameters of pigs. This could enhance our understanding of the infection mechanisms and aid in developing targeted treatment strategies.Evaluation of the effectiveness of antibiotics: Respondents would also like to see a comprehensive investigation into the effectiveness of antibiotics in combating chlamydia-related fertility problems. This could help to optimise the treatment duration and dosage of antibiotic therapy and identify alternative treatment options.

## 4. Discussion

Since information on the individual veterinarian has not been collected, a clustering effect of veterinarians working in the same practice cannot be evaluated but might be present. Moreover, there may be regional variations in chlamydia prevalence that influence decisions of veterinarians regarding diagnostics, treatment, and prevention, which must be considered. However, most pig practitioners in Austria generally work on a cross-regional basis, assuming that the impact of personal experience related to chlamydiosis is greater than that of regionality. Indeed, there is a considerable divergence of opinions on what clinical chlamydial signs of disease in sows could potentially look like. “Rebreeding” was the most mentioned clinical observation associated with chlamydia. Diagnostic strategies in the case of rebreeding are also not easy and very diverse, since besides chlamydia and other infectious diseases, numerous non-infectious causes (stress, malnutrition, barn temperatures, incorrect insemination procedures, and more) [11] could be involved. For chlamydia, the key issue in the interpretation of results for direct detection is that there are no official sampling guidelines based on the fact that there are no experimental infection studies in sows to recommend herd sampling strategies in case of fertility problems. It is therefore up to each individual veterinarian to decide how and when samples are collected, and which method is used for testing. Vaginal swabs might not be the best choice for sampling material, because the anatomical proximity of the vagina and anus allows chlamydia, which is regularly excreted in faeces, to potentially ascend into the vagina without necessarily causing problems. Cervical swabs might therefore be more informative. Kauffold et al. (2006) were among the first examining the porcine uterus. On the one hand, they were successful in proving chlamydia is involved in ultrastructural changes in the oviduct of sows; on the other hand, they could not show any correlation between histological results and direct detection by PCR and/or immunohistochemistry [12]. Polish researchers performed statistical analyses of fertility parameters in herds with and without *C. suis* infections and could show that the type of parameter (abortion or rebreeding) is not an impact factor for chlamydiosis [13,14]. Due to the lack of infection studies in pregnant sows, the ideal foetal tissue for detecting chlamydia in the case of abortion remains unclear. Identification of chlamydia in porcine abortion materials has been documented in a few case reports since 1997 [3,7,15]. In this respect, it is important to state that no experimental setting has been able to induce abortion in pregnant sows so far [16]. The association of chlamydia with mummification originates from the study conducted by Plagemann (1981), who was able to detect elementary bodies in the gastric mucosa of mummified foetuses, but also in non-mummified littermates [5]. Chlamydia was identified in twenty cases of weak-born piglets by ELISA and/or electron microscopy, mostly in combination with other pathogens [6]. Eggemann (2000) found a statistically significant higher number of chlamydia PCR positive cervical swabs in sows with vaginal discharge, abortion, rebreeding, and weak-born piglets compared to sows without these issues [2]. To date, intrauterine or placental transmission has also not been confirmed. Positive PCR detection could therefore also be due to contamination by faecal chlamydia or chlamydia from the lower genital tract.

In humans, both the World Health Organization (WHO) and Centers for Disease Control and Prevention (CDC) recommend either macrolide azithromycin (1 g orally as a single dose) or doxycycline (100 mg orally twice a day for seven days) as first-line treatments for chlamydia [17,18]. Alternatively, 500 mg of tetracycline four times a day for seven days can be given [18]. Both azithromycin and tetracyclines bind to the 50S subunit of the 70S bacterial ribosomes and inhibit the protein synthesis in chlamydia, but in contrast to tetracyclines, azithromycin is not registered for use in pigs. Nevertheless, other registered macrolides could be used, but no data about dosage and treatment length exist for their usage in pigs in case of chlamydiosis. Most practitioners treat the sow herd orally via feed with tetracyclines, and about 50% stated treatment lengths between 10 and 15 days. This treatment strategy seems to have positive effects on the gestation rate; at least, this can be concluded from the veterinarians’ responses and repeated application not only for treatment but also prevention. Nonetheless, practitioners would appreciate comprehensive investigations into the effectiveness of this therapeutic approach.

Awareness of high prevalence rates of tetracycline resistance in *C. suis* isolates seems to be low among veterinary practitioners but can be easily explained by the fact that hardly anybody tests for tetracycline resistance in routine diagnostics. This makes transparent communication between science and practitioners extremely important; at the same time, the textbooks claiming tetracyclines as the treatment of choice need to be revised and updated accordingly. While in humans treatment and preventing strategies always include the sexual partner [17], sows are routinely inseminated with semen obtained from boars in official boar studs, all tested for chlamydia on a regular base. Semen does not seem to be the key factor in transmission in sow herds, but rather the insemination technique, as chlamydia is rectally shed [19] and might get into the genital tract. Mice, rats, birds, or cats could also play a role in transmission, but evidence is anecdotal only. It was shown that chlamydia does not survive in dust [20]; therefore, it might be important to keep the environment dry. There are still gaps in understanding the antichlamydial effect of disinfectants, one of the main questions requested by the practitioners.

## 5. Conclusions

The practitioners’ assessments of clinical occurrence, diagnostic procedures, interpretation of lab results, treatment of sows, and prophylactic procedures are extremely diverse. Recommendations from the scientific community based on experimental evidence are needed. This would be very important to provide practitioners with guidelines and facilitate decision-making with regard to treatment and prophylaxis. This would not only be important for providing adequate veterinary care to the sows but also in terms of reducing disease burden overall, reducing zoonotic risk as part the One Health principle.

## Figures and Tables

**Figure 1 animals-14-02983-f001:**
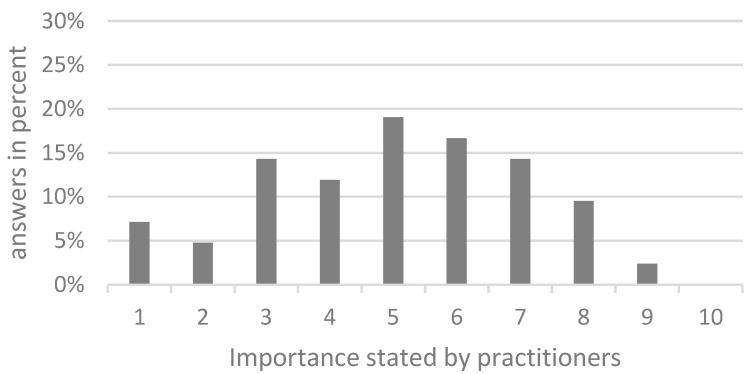
Rating of the general importance of chlamydial infections in the context of fertility problems on a Likert scale from one to ten (1 = not important; 10 = very important) as stated by the respondents (n = 42).

**Figure 2 animals-14-02983-f002:**
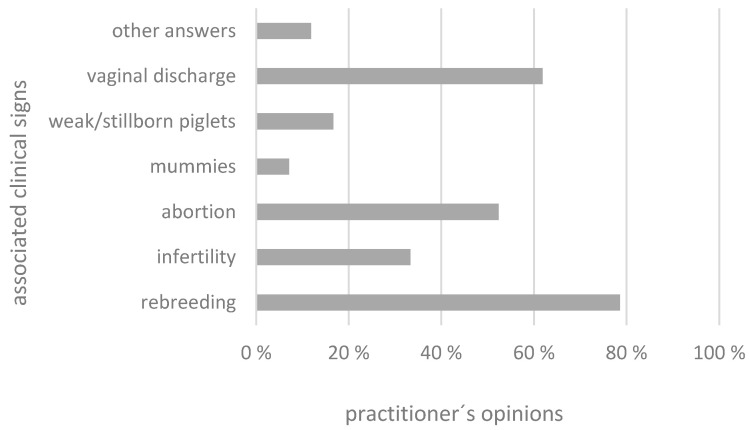
Clinical signs associated with chlamydia-induced fertility problems in sows named by the respondents in percentages.

**Figure 3 animals-14-02983-f003:**
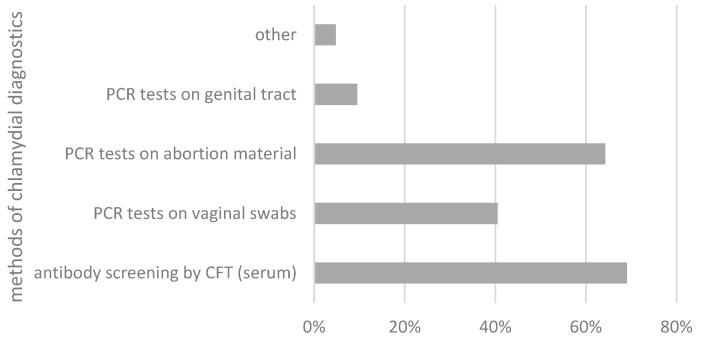
Methods chosen by practitioners for the diagnosis of chlamydia-associated fertility problems in percentages. CFT: complement fixation test; PCR: polymerase chain reaction.

**Figure 4 animals-14-02983-f004:**
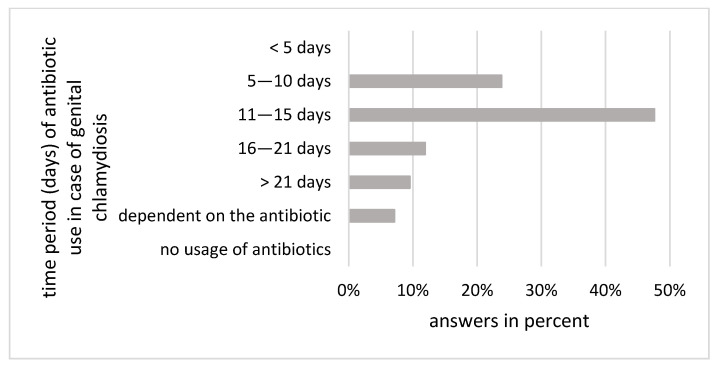
Duration of antibiotic treatment (mainly tetracyclines) in case of suspected genital chlamydiosis according to practitioners in percentages.

**Table 1 animals-14-02983-t001:** List of prophylactic measures performed by swine practitioners in order to avoid chlamydial infections (in percentages). The percentages of the sub-terms always refer to the data of the general terms.

Prophylactic Measures		Answers
Hygiene		57%
	Stable hygiene	27.8%
	Cleaning and disinfection	25.0%
	Rodent control	19.4%
	Insemination management	13.9%
	Bird control	13.9%
Antibiotics (tetracyclines)		41%
	Whole-herd-based (1–2 times/year)	23.5%
	Production-stage-based (time around insemination)	47.0%
	Gilts/quarantine	17.6%
	No further details mentioned by the veterinarians	11.9%
Supporting reproductive performance		21%
	Use of hormones	22.2%
	Vaginal/uterine lavages	55.6%
	Exchange of boar or artificial insemination	22.2%

## Data Availability

Data are contained within the article.

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
