# Peer review of "Suspected Cases of Chlamydia-Induced Fertility Problems in Sows: What Is the Approach of Austrian Practitioners?"

_animals, 2024, doi:10.3390/ani14202983_

Round 1
Reviewer 1 Report
Comments and Suggestions for Authors
Unterweger and Buzarnich-Ladinig have presented findings of a survey of Austrian veterinary practitioners on their experiences relating to cases of chlamydia-induced infertility. The survey provides valuable insights into the diagnostic tests and treatments used by some of the veterinarians responsible for some of the largest sow farms in Austria. What is clear from the study is that a concerted effort by Scientists working on swine chlamydial infections need to interact and advise more closely with veterinarians to come up with a consensus pathway for diagnosis and treatment of C. suis suspected cases.
Some specific comments to authors are listed below:
A good manuscript encapsulating valuable opinions and trends of sow veterinarians in diagnosis and treatment of C. suis suspected cases. Specific comments to address improve readability of the manuscript are as follows:
Simple abstract
Line 11: Change the word ‘overlook’ to ‘monitor’.
Line 20: change the sentence to read ‘…due to the lack of scientifically validated guidance on control strategies including treatment lengthy, treatment dose and disinfection procedures.’
Introduction
Line 49: change ‘animal experiments’ to C. suis in vivo studies..’
Line 50: change ’choice’ to ‘selection of appropriate sampling material….’
Line 61: change ‘application form’ to ‘delivery route’
Line 63-64: Are methods developed for the control of other chlamydial species not effective for C. suis. Please expand on this point.
Line 66: This is incorrect the way this is written about there being no chlamydial vaccines. Please rephrase this statement to state that vaccines exist for a few chlamydial species namely C. abortus and C. felis but not yet for C. suis.
Line 68: change ‘base and having an obligation…’ to ‘basis and have , therefore an obligation..’
Line 69: Please insert the text ‘of this study’ after ‘The answers…’
Line 70: change ‘fields of research’ to ‘areas of investigation with the main aim in the prevention of chlamydiosis in sows’.
Materials and methods
Line 75: word distributed is spelt incorrectly. Change the word ‘per’ to ‘by’.
Results
Lines 100-153: Please be consistent when you mention ‘percentages’ and ‘n=’, either keep these together in a single set of brackets or keep separated in two sets of brackets.
Line 115: Please explain more fully what is meant by the term ‘rebreeding’ as indicated in figure 2 and the questionnaire.
Figure 3 legend: Please state the full name of the techniques listed ie CFT and PCR.
Line 150: Please rephrase the sentence ‘No one chose not to use…’ to ‘All used antibiotics as a treatment option…’
Line 153: change the word ‘treats’ to ‘treated’
Table 1: Please clarify the last option under the antibiotics section ‘nur further details 11,9%.’
Line 193: Insert the word ‘early after ‘effective’.
Discussion
Line 214 change to read ‘opinions on what clinical chlamydial signs of disease in sows…’
Line 218: change ‘unproper’ to ‘incorrect’.
Line 262: Please amend sentence to read ‘Awareness of high prevalence rates of tetracycline resistance in C. suis isolates seem to be low among veterinary practitioners, but can….’
Line 266: change the ending ‘rewritten.’ to ‘revised and updated accordingly.’
Line 272: Please reword, but was never proven’ to something along the lines of ‘evidence is anecdotal only…’
Line 273-4: Please change to ‘gaps in understanding’ from ‘studies missing’.
Conclusions
Line 279: insert ‘based on experimental evidence are needed.’ after ‘the scientific community…’. Then add the text ‘This would be important…..’
‘based on experimental evidence’
Line 282 add text ‘reducing disease burden overall reducing zoonotic risk as part…’ after ‘in terms of…’
Comments on the Quality of English LanguageThe manuscript generally reads well. There are minor typographical changes recommended and these are stated in the above comments, section by section.
Author Response
Response to Reviewer 1:
- Summary:
Unterweger and Buzanich-Ladinig have presented findings of a survey of Austrian veterinary practitioners on their experiences relating to cases of chlamydia-induced infertility. The survey provides valuable insights into the diagnostic tests and treatments used by some of the veterinarians responsible for some of the largest sow farms in Austria. What is clear from the study is that a concerted effort by Scientists working on swine chlamydial infections need to interact and advise more closely with veterinarians to come up with a consensus pathway for diagnosis and treatment of C. suis suspected cases.
We sincerely thank the reviewer for the thorough evaluation and constructive feedback, especially for the suggestions of wordings. We have carefully addressed each comment, making substantial revisions to improve clarity and detail, as well as clarifying any uncertainties. We hope that these revisions sufficiently address your idea.
- Simple abstract
Line 11: Change the word ‘overlook’ to ‘monitor’. The wording was changed – (line 11)
Line 20: change the sentence to read ‘…due to the lack of scientifically validated guidance on control strategies including treatment lengthy, treatment dose and disinfection procedures.’ The wording was changed – (lines 20-21)
- Introduction
Line 49: change ‘animal experiments’ to C. suis in vivo studies..’ The wording was changed – (line 49)
Line 50: change ’choice’ to ‘selection of appropriate sampling material….’ The wording was changed – (lines 50-51)
Line 61: change ‘application form’ to ‘delivery route’ The wording was changed – (line 61)
Line 63-64: Are methods developed for the control of other chlamydial species not effective for C. suis. Please expand on this point. We thank you for this important comment. We added the following sentences (lines 64-68): “Methods developed for controlling other chlamydial species may not be fully effective for Chlamydia suis, on the one hand due to the aforementioned tetracycline resistance and antibiotic regulations in livestock medicine, and on the other hand due to differences in the husbandry of herd animals in barns, usually associated with a high animal density compared to pets kept individually.”
Line 66: This is incorrect the way this is written about there being no chlamydial vaccines. Please rephrase this statement to state that vaccines exist for a few chlamydial species namely C. abortus and C. felis but not yet for C. suis. Thank you for pointing this out. We added this information in lines 70-72: “Finally, unlike C.abortus vaccines in ruminants or C. felis vaccines in cats, vaccines against C.suis induced clinical problems in pigs are not available.”
Line 68: change ‘base and having an obligation…’ to ‘basis and have , therefore an obligation..’ The wording was changed – (line 74)
Line 69: Please insert the text ‘of this study’ after ‘The answers…’ The wording was changed – (line 75)
Line 70: change ‘fields of research’ to ‘areas of investigation with the main aim in the prevention of chlamydiosis in sows’. The wording was changed – (lines 76-77)
- Materials and methods
Line 75: word distributed is spelt incorrectly. Change the word ‘per’ to ‘by’. The wording was changed – (line 85)
- Results
Lines 100-153: Please be consistent when you mention ‘percentages’ and ‘n=’, either keep these together in a single set of brackets or keep separated in two sets of brackets.
The authors thank you for this comment. Percentages and numbers were standardised (percentage %; n=x).
Line 115: Please explain more fully what is meant by the term ‘rebreeding’ as indicated in figure 2 and the questionnaire.
We agree with this comment that it is necessary to add this information. An explanation was added in lines 120-121: “meaning that previous mating attempts did not succeed”
Figure 3 legend: Please state the full name of the techniques listed ie CFT and PCR.
Thank you for this comment. We added the full names of CFT and PCR (lines 150-152)
Line 150: Please rephrase the sentence ‘No one chose not to use…’ to ‘All used antibiotics as a treatment option…’ The wording was changed – (line 157)
Line 153: change the word ‘treats’ to ‘treated’ The wording was changed – (line 160)
Table 1: Please clarify the last option under the antibiotics section ‘nur further details 11,9%.’
Thank you for this comment. This is a spelling mistake and is supposed to mean “no further details”. We changed the wording to “No further details mentioned by the veterinarians”
Line 193: Insert the word ‘early after ‘effective’. The word was inserted (line 199)
- Discussion
Line 214 change to read ‘opinions on what clinical chlamydial signs of disease in sows…’ The wording was changed – (line 227)
Line 218: change ‘unproper’ to ‘incorrect’. The wording was changed – (line 231)
Line 262: Please amend sentence to read ‘Awareness of high prevalence rates of tetracycline resistance in C. suis isolates seem to be low among veterinary practitioners, but can….’ The wording was changed – (line 275)
Line 266: change the ending ‘rewritten.’ to ‘revised and updated accordingly.’ The wording was changed – (line 279)
Line 272: Please reword, but was never proven’ to something along the lines of ‘evidence is anecdotal only…’ The wording was changed – (line 286)
Line 273-4: Please change to ‘gaps in understanding’ from ‘studies missing’. The wording was changed – (line 288)
- Conclusions
Line 279: insert ‘based on experimental evidence are needed.’ after ‘the scientific community…’. Then add the text ‘This would be important…..’ ‘based on experimental evidence’ The wording was inserted – (lines 293)
Line 282 add text ‘reducing disease burden overall reducing zoonotic risk as part…’ after ‘in terms of…’ The text was added – (lines 296-297)
Reviewer 2 Report
Comments and Suggestions for Authors
Nice paper. As expected a high degree of variation in opinion and approach on a a subject that we know very little about.
The only comment I have is that you state that the respondents were from the largest pig practices. Because you did not collect any identifying information, how did you deal with the potential "culstering effects" of vets from the same practice having similar ideas due to discussions amongst the vets in that practice?
What about regional effects? You state that experience with C suis played a key role in how they viewed the issue, so one could assume that if one region saw more suspected C suis then there might be different views based on prevalence?
I think addressing those issues, even in the discussion, would improve the paper.
Author Response
Response to Reviewer 2:
Nice paper. As expected a high degree of variation in opinion and approach on a a subject that we know very little about. The only comment I have is that you state that the respondents were from the largest pig practices. Because you did not collect any identifying information, how did you deal with the potential "clustering effects" of vets from the same practice having similar ideas due to discussions amongst the vets in that practice?What about regional effects? You state that experience with C suis played a key role in how they viewed the issue, so one could assume that if one region saw more suspected C suis then there might be different views based on prevalence?I think addressing those issues, even in the discussion, would improve the paper.
We sincerely thank the reviewer for the thorough evaluation and constructive feedback! Your comment is fully justified, and we added some sentences arguing this issue. You will find them at the beginning of the discussion (lines 220-226). We think they fit well, and hope that you are satisfied by their content: “Since information on the individual veterinarian has not been collected, a clustering effect of veterinarians working in the same practice cannot be evaluated but might be present. Moreover, there may be regional variations in chlamydia prevalence that influence decisions of veterinarians regarding diagnostics, treatment, and prevention which must be considered. However, most pig practitioners in Austria generally work on a cross-regional basis, assuming that the impact of personal experience related to chlamydiosis is greater than that of regionality.”